# A High-Throughput and Uniform Amplification Method for Cell Spheroids

**DOI:** 10.3390/mi13101645

**Published:** 2022-09-30

**Authors:** Liyuan Liu, Haixia Liu, Xiaowen Huang, Xiaoli Liu, Chengyun Zheng

**Affiliations:** 1Department of Hematology, The Second Hospital, Cheeloo College of Medicine, Shandong University, Jinan 250033, China; 2State Key Laboratory of Biobased Material and Green Papermaking, Department of Bioengineering, Qilu University of Technology (Shandong Academy of Sciences), Jinan 250300, China; 3Department of Reproductive Medicine, The Second Hospital, Cheeloo College of Medicine, Shandong University, Jinan 250033, China

**Keywords:** cell culture, 3D cell spheroids, high-throughput, uniform shape, drug screening

## Abstract

Cell culture is an important life science technology. Compared with the traditional two-dimensional cell culture, three-dimensional cell culture can simulate the natural environment and structure specificity of cell growth in vivo. As such, it has become a research hotspot. The existing three-dimensional cell culture techniques include the hanging drop method, spinner flask method, etc., making it difficult to ensure uniform morphology of the obtained cell spheroids while performing high-throughput. Here, we report a method for amplifying cell spheroids with the advantages of quickly enlarging the culture scale and obtaining cell spheroids with uniform morphology and a survival rate of over 95%. Technically, it is easy to operate and convenient to change substances. These results indicate that this method has the potential to become a promising approach for cell–cell, cell–stroma, cell–organ mutual interaction research, tissue engineering, and anti-cancer drug screening.

## 1. Introduction

The traditional two-dimensional (2D) cell culture technology was created more than a century ago. Since then, with the rapid development of science and technology, cell culture has become a critical life science research technology applied in life science, tissue engineering, and medical research [1,2]. People make cells grow and expand in vitro environments by simulating the in vivo environment and use this process to carry out cell anabolism [3], signal transduction, and application research [4,5].

The traditional 2D cell culture is usually carried out on a porous plate, a Petri dish, and a tissue culture flask made of glass or polystyrene. The cultured cells are in a monolayer state [6]. The method is most common in cell culture and has the advantages of simple operation, low cost, stability, high cell activity, and suitability for multiple cell types [6,7]. In past research, 2D cell culture technology has significantly improved the understanding of cell structure and development research. However, with the exploration, the shortcomings and limitations of 2D cell culture technology gradually emerged. Scientific research has found that 2D cell culture technology cannot simulate the living environment of normal cells in vivo. In addition, the application of 2D cell culture technology will reduce the multi-lineage differentiation potential of biological cells on the one hand [8] and change the phenotype of biological cells on the other hand [9].

Therefore, in line with the development needs of life science, three-dimensional (3D) cell culture technology has been widely promoted.

Unlike traditional 2D cell monolayer culture technology, 3D cell culture technology simulates the natural cell environment of growth and structural specificity in vivo. Li et al. found that cells growing in 3D spheres exhibited higher proliferative capacity and stronger cell-to-cell adhesion [10]; Chaicaroenaudomrung et al. found that 3D cell culture technology helps to study the mutual interaction of cell–cell, cell–stroma, and cell–organ [11]. This type of technology could provide more accurate information for tumors or disease treatment, characterization of tissue and organs [12], metabolic analysis [13], in vitro drug screening, and stem cell research [14,15,16].

Nowadays, people have developed several 3D cell culture methods after numerous explorations. Examples include the hanging drop method [17,18], the spinner flasks method [19], the ultra-low attachment plate culture method [20], the microbead method [21], the micropatterned method [22], and so on.

However, these methods have some disadvantages. For example, in the hanging drop method, on the one hand, the cell growth space is small. On the other hand, due to the inverted state in the culture process, it is inconvenient to change the culture medium, administer drugs, and perform other operations [23]. However, the medium in this method evaporates quickly with time, so it is essential to change the medium in time. Additionally, the vitality of the cultured cells by this method is low [24]. These reasons make the hanging drop method unsuitable for long-period cell culture.

In the spinner flask method, the stirring rate is required to a certain extent, and an excessive stirring rate may cause mechanical damage to the cells. However, a slow stirring speed will lead to the cell sinking to the bottom of the container, resulting in an adherent phenomenon [23]. The ultra-low attachment plate culture method is challenging to control the culture uniformity of cell spheres, and the sizes of the cultured cell spheroids are different to a certain extent [8]. The micropatterned method requires special chemical technology and photosensitive materials to prepare micropattern structures, which are complicated to operate and has poor biocompatibility [25].

To solve the problems mentioned above, many explorations have been made. Mary et al. designed a microfabricated magnetic pattern to achieve high-throughput amplification of EBs [26]. Zhao et al. invented a 3D printed hanging drop dripper, simplifying the manufacturing process of micromechanical systems [27]. Su et al. reported a simple and expansile ECM patterning method for microfluidic that is easy to apply in closed microchannels or exposed outer surfaces and very suitable for in-gel cell culture [28]. Shao et al. constructed a new layered inverse opal porous scaffold and realized the large-volume cell culture [29]. However, it is still challenging for these 3D cell culture technologies to create high-throughput and ensure uniformity of the cell spheres.

Here, we report a novel cell spheres culture method that can achieve high-throughput and uniform cell spheres amplification. This method is simple and easy to operate, and it is convenient to replace and add materials to the cell culture system. As shown in Figure 1, soft lithography prepared a polydimethylsiloxane (PDMS) chip with a uniform array of micropores. We added a specific concentration of cell suspension and culture medium into the microwell chip (Figure 1a). After the cells settled into the micropores of the chip (Figure 1b), the excess cells on the chip surface were slowly washed away by the culture medium (Figure 1c). Then, a sufficient volume of culture medium was added to cover the surface of the chip. The treated chip was transferred to an incubator at 37 °C for culture. After the cells in the micropore spontaneously gathered into uniform-sized cell spheres (Figure 1d), the microwell chip was inverted (Figure 1e) to separate the cell spheres in the micropore. Then, the isolated cell spheres were transferred to stirred culture flask, which was coated with polydimethylsiloxane (PDMS) on the inner walls (Figure 1f). Subsequently, high-throughput uniform amplification of the cell spheres was performed under appropriate conditions (Figure 1g). The cell spheres’ morphology and survival were observed and recorded during the culture.

In the common microwell chip culture method [8,30,31], the cell spheroids’ size is determined by conditions such as culture time and microwell size. The cell spheroids’ shape obtained by too long or too short culture time will be somewhat different, and the microwell size is not easy to change casually.

However, in this method, we can control the size of the obtained cell spheroid by controlling the rotation speed of the magnetic stirrer, and this method has the following advantages: (1) When washing the cells on the surface of the chip, the cells settled in the micropores are not easy to be taken out, thus avoiding repeated inoculation and simplifying the steps; (2) When adding culture medium and administration factors during amplification culture, the cell spheroids will not be taken out of micropores to form irregular cell aggregates, improving the utilization rate of cells and avoiding the interference to the application results of cell spheroids; (3) Good repeatability—the chips and the stirred culture flask used in this method can be easily cleaned and sterilized; (4) The single-layer chip used in this experiment can initially form 841 cell spheroids at one time. And the microwell chip and the stirred culture flask can be enlarged as required. The excellent biocompatibility and hydrophobicity of PDMS inhibit the phenomenon of cell adhesion so that high-throughput cell spheroids can be obtained by one-time culture.

Apart from the above advantages, the size of the array micropore on the chip is uniform, and the stirring speed is controllable. The obtained cell spheres are uniform in morphology and can be used in high-throughput drug screening, which is easy to observe. The cultured cell sphere is closer to the perfect cell sphere, which is beneficial to the study of the oxygen microenvironment of the multicellular tumor sphere [32]. It also has the advantages of convenient material replacement and addition, simple operation, and no use for complicated chemical technology and photosensitive materials. In addition, the survival rate of this method is over 95%, which is suitable for many kinds of cells. The method is conducive to establishing the 3D cell spheroid model in vitro for drug analysis and improving drug development efficiency. The method simulates the living environment of normal cells in vivo, retains the good multi-directional differentiation potential of biological cells, and does not change the phenotype of biological cells. We believe this method may be one of the most potent ways to study cell–cell, cell–stroma, and cell–organ mutual interactions in the future.

## 2. Materials and Methods

### 2.1. Preparation of Microwell Chip and Stirring Culture Flask

#### 2.1.1. Materials

The PDMS brand model used was RTV615 of Momentive, USA. The flask body and cap material was a low boron silicon pharmaceutical special glass flask and polypropylene (purchased from Taobao). The stirring support, stirring shaft, and fixing clip material was nylon 12 (obtained by 3D printing), and the thin metal stick material was stainless steel (purchased from Taobao).

#### 2.1.2. Preparation of Microwell Chips

The chip was made by soft lithography, and its size was 20 mm × 16 mm × 2.5 mm (Length × width × height); There were 841 (29 × 29) micropores (d = 0.2 mm, volume = 1.26 × 10^−3^ mm^3^) uniformly arranged on the same chip (Figure 1h,i). A certain proportion (essential component:curing agent = 10:1) of PDMS prepolymer was transferred onto a silicon wafer with a uniform micropore array and placed in a vacuum dryer for vacuuming for 10 min, taken it out, and removed air bubbles with an ear washing ball. After the PDMS was uniformly paved, it was transferred to an oven at 80 °C for complete curing. Then, the solid, entirely cured PDMS was separated, and the microwell chip could be obtained by cutting. The size and arrangement of the micropores array could be changed according to the design of the silicon chip.

#### 2.1.3. Preparation of Stirring Culture Flask

The stirring culture flask body was in a cylindrical flask structure (Figure 1j), and the inner wall of the flask was coated with a layer of dried hydrophobic PDMS (essential component:curing agent = 10:1). The flask cap was in a thread structure, and enough area was reserved for connecting the inner side of the flask cap with the stirring support. The bottom of the stirring support was designed with a hole for the stirring shaft to penetrate through. A circular hole with a diameter of 1 mm was reserved at the upper end of the stirring shaft for the thin metal rod to penetrate through. The stirring shaft was connected with the stirring support so that the stirring shaft could not fall off from the stirring support while rotating. Both ends of that fixed clamp were seal designs to prevent the magnetic stirring bar from radially falling off in the rotation process.

The assembly process was as follows: firstly, the magnetic stirring bars were clamped into the fixing clamp to ensure that the magnetic stirring bar was fixed in good condition. Then the upper end of the stirring shaft passed through the stirring support, and the thin metal stick symmetrically passed through the small hole at the upper end of the stirring shaft. After that, the upper end of the stirring support of the stirring shaft was coated with special polypropylene glue so that the stirring support was fixed at the center of the inner side of the flask cap and dried in an oven at 80 °C for 3 h to ensure firm bonding. During this period, PDMS was prepared according to the proportion (essential component:curing agent = 10:1), then evenly coated on the inner flask wall, and dried upside down in an oven at 80 °C for 30 min. Finally, the processed stirring component and the flask body were assembled into an integrated stirring culture flask through the flask cap.

### 2.2. Cell Culture

The synovial sarcoma cell line HS-SY-II (obtained from Dr. Changliang Peng) and human umbilical cord-derived mesenchymal stem cells (UC-MSCs) were utilized in this experiment. HS-SY-II cells were cultured in Dulbecco’s Modified Eagle Medium (DMEM), and MSCs were cultured in DMEM/F12 medium. All media were supplemented with 0.1 mg/mL streptomycin sulfate, 100 U/mL penicillin (Life Technologies, Carlsbad, CA, USA), and 10% fetal bovine serum (FBS). Both cell lines were incubated at 37 °C in a humidified atmosphere with 5% CO_2_.

### 2.3. Cell Spheroids Formation

First, we put the chip into the Petri dish so that the bottom of the chip was in close contact with the Petri dish to prevent the chip from separating from the Petri dish after adding the medium. Second, we added an appropriate amount of cell culture medium to the chip and then put the Petri dish into a vacuum-drying oven until the air bubbles in the inner wells of the chip were released. Then, we sucked out the air bubbles with a pipette and refilled the medium in a volume just enough to penetrate the holes on the chip.

HS-SY-II and MSC cells were digested with 0.05% trypsin-EDTA (Invitrogen, Carlsbad, CA, USA) and resuspended in the culture medium. After that, the cell suspension was formulated at a concentration of 1 × 10^7^ cells/mL using the cell culture medium. About 30 μL of the suspension was introduced onto the chip. After about 4 min, we rinsed the excess cells on the chip with the culture medium to allow the cells to sink into the micropores. It was necessary to rinse slowly and repeatedly with a 10 μL pipette to prevent flushing out cells located in the micropores. When it was observed under the microscope that the micropores of the chip were full of cells and there were not too many extra cells on the chip, a sufficient volume of cell culture medium was added to the dish to cover the chip.

After one day of culture in a 37 °C incubator, cells spontaneously aggregated into spheroids. The next day, the spheroids were transferred into the stirred culture flasks, which were placed on a magnetic stirrer (IKA, color squid), and a five-day culture was maintained. Incubate the spheroids at 37 °C for five days in a carbon dioxide incubator. The morphology of the spheroids and the changes in living and dead cells were observed and photographed using the inverted fluorescence microscope and the optical microscope every day.

### 2.4. Cell Viability Test

To assess the viability of the spheroids, we stained some spheroids daily with Calcein AM and propidium iodide (PI) during the culture in the flask. After pipetting several times, we aspirated about 2 mL of the medium in the flask daily and transferred it to a new culture plate (48 wells). After three washes with 1X Assay Buffer, the spheroids were suspended in 2 µm Calcein AM and 4.5 µm PI solution and incubated at 37 °C for 30 min. Then, we imaged it with Leica DMi8 inverted fluorescence microscope (Leica Microsystems, Wetzlar, Germany).

## 3. Results

### 3.1. 3D Spheroid Formation and Maintenance

About 30 μL of the cell suspension, whose concentration was 1 × 10^7^ cells/mL, was introduced onto the microwell chip. Due to gravity, cells sank into the micropores in the chip to form aggregates and adhered to each other (Figure 2a (Day 0) and Figure 3 (Day 0)). The number of cells settled in each micropore is shown in Figure 2b. After 24 h, The spheroids formed in the micropores were observed (Figure 2a (Day 1) and Figure 3 (Day 1)). After 48 h, the spheroids’ surface became smooth, which indicated that the cells had firmly aggregated to form tight junctions (Figure 4a (Day 2) and Figure 5a (Day 2)). Subsequently, the spheroids were transferred to the flasks, which were coated with PDMS. Later, from Day 2 to Day 6, there was no significant change in the volume of the cell spheroids (Figure 4a and Figure 5a). Overall, this process consisted of two stages: (I) Cells aggregated into spheroids within 24 h of incubation in chip micropores; (II) After 24 h of seeding on the chip, the morphology and viability of the spheroids were maintained in the stirred flask for another five days.

### 3.2. Cell Viability

During the culture in the flask, from Day 2 to Day 6, about 2 mL of liquid was taken from the flask, and the spheroids were stained with Calcein AM and propidium iodide (PI). Stained spheroids were imaged using a Leica DMi8 inverted fluorescence microscope. The staining results showed very high viability during the six-day culture, and the quantitative analysis indicated that the viability of the cell spheroids was above 95% (Figure 4b and Figure 5b).

## 4. Discussion

The micropores culture in the chip presented in this study resulted in a high-throughput, uniform shape of the formed spheroids and high viability of the formed cell spheroids. Subsequently, the formed cell spheroids with uniform size and a large number were transferred to the flask with a magnetic stirrer for further expansion of the cultured cell spheroids. Finally, the cell spheroids were cultured in the flask until the 5th day, and the cell viability remained above 95%.

In the previous work [25], we found that compact cell spheroids could be formed on the 1st day of cultivation, and these cell spheroids gradually became larger on the 2nd and 6th days. On the seventh and eighth days, their size did not change significantly, but dead cells were observed on the surface of spheroids. On the ninth day, many cells detached from the spheroids, and the size of aggregates decreased slightly. In future research, we will further improve the culture system by exploring the rotation speed of the stirring culture flask, the culture temperature, and the improvement of the structure of the flask so that the expansion or enlargement of the spheres is more obvious. We will continue to explore and compare the similarities and differences in cell properties between the microwell chips and stirring culture flask system used in this article and the previous ordinary 2D culture.

Further exploration of this system in the rapid and massive expansion of stem cells and the maintenance of stem cell stemness and other properties provides hope for the large-scale expansion and maintenance of various stem cells. We will explore the application of different types of cells in this system as much as possible and further expand the research of this equipment system in drug screening and other aspects.

PDMS is the preparation material of the chip, and it is also the coating material of the inner wall of the stirred culture flask. In this study, we make full use of the advantages of PDMS that its hydrophobicity is unfavorable for cell adhesion and its excellent biocompatibility. Compared with other materials, the material used in our research is more conducive to high-throughput culture and amplification of cell spheres which shows overwhelming advantages over convenient operations.

## 5. Conclusions

We applied an easy-to-manipulate and beneficial micropores culture chip to form 3D spheroids of cells. After the cell suspension was pipetted into the chip, under an optical microscope, we could observe the cells aggregated in the micropores and formed 3D cell spheroids within 24 h. During the seven-day culture and observation period, the formation and evolution of 3D cell spheroids mainly experienced the following stages: Initially, the cells in the suspension sank into the micropores of the chip by gravity. Over time, the cells in the micropores aggregated into relatively loose cell spheroids. Then, the spheroids gradually became compact, and the surfaces of the spheroids became smooth. 3D cell viability analysis shows that the viability of the cell spheres formed on the chip and transferred to the flask for further culture is mostly above 95%. These experimental results indicate that the combined techniques of forming cultured cell spheroids on our newly designed chip and moving the cell spheroids to a flask and placing them on a magnetic stirrer have obvious advantages, which highlight a great potential application of this method in the field of cell research and cell industry in the future.

## Figures and Tables

**Figure 1 micromachines-13-01645-f001:**
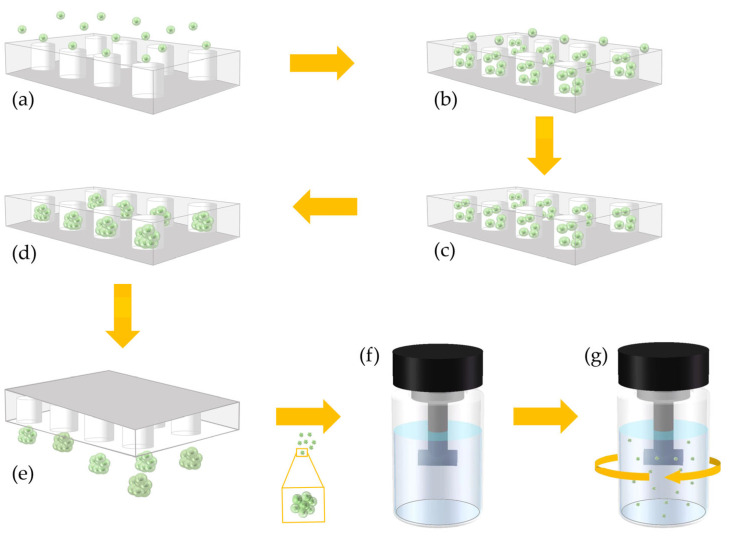
Schematic Diagram of High-Throughput and Uniform Cell Sphere Amplification Method. (**a**) Introducing cells into a microwell chip; (**b**) cells settle into micropores; (**c**) washing out the redundant cells on the surface of the microwell chip; (**d**) the cells in the micropores are aggregated into cell spheres; (**e**) inverting the microwell chip and taking out the cultured cell spheres; (**f**) introducing the cell spheres taken out of the microwell chip into a stirring amplification device; (**g**) dynamically amplifying the cell spheres in a stirring amplification device; (**h**) microwell structure. D = 200 μm, magnification = 40×, bar = 400 μm; (**i**) Photographs of the microwell chip, bar = 5000 μm; (**j**) stirring the amplification device. Bar = 10,000 μm.

**Figure 2 micromachines-13-01645-f002:**
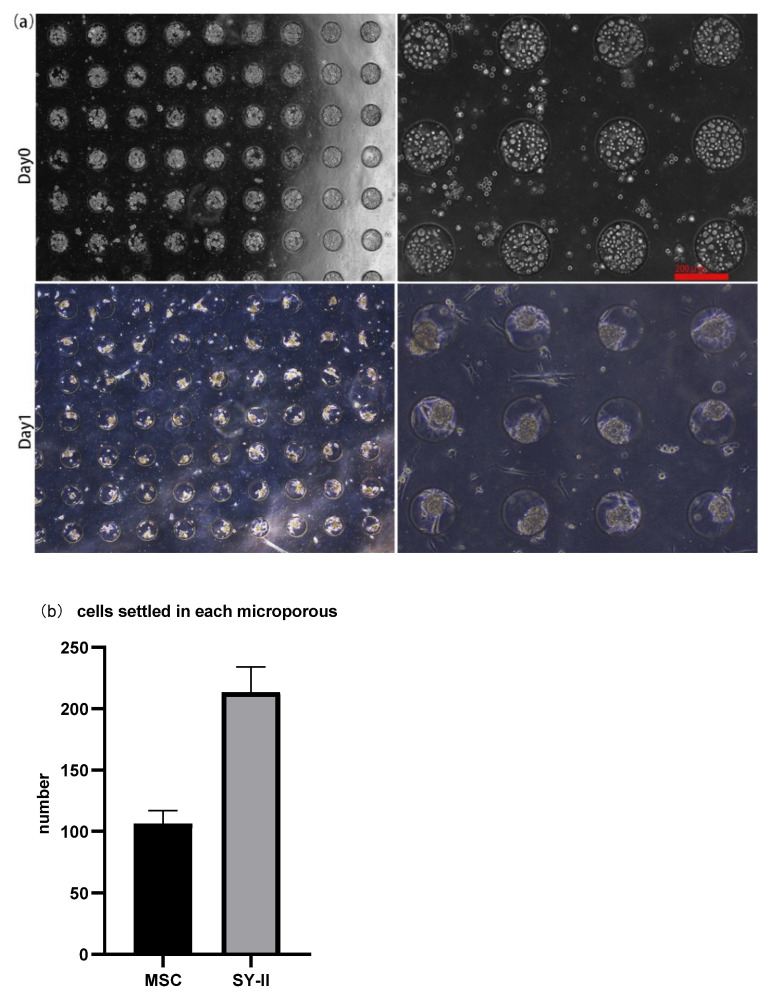
(**a**) The morphology of MSC cells on the day of seeding (Day 0) and spheroid formation (Day 1). On day 0, about 30 μL cell suspension at a concentration of 1 × 10^7^ cells/mL was added to the chip. On day 1, 3D spheroid formation occurred. Magnification = 40× (two pictures on the **left**); Magnification = 100× (two pictures on the **right**). Bar = 200 μm. (**b**) Histograms showed the number of cells that settled into each micropore and eventually formed spheroids of two cell lines, MSC and SYII.

**Figure 3 micromachines-13-01645-f003:**
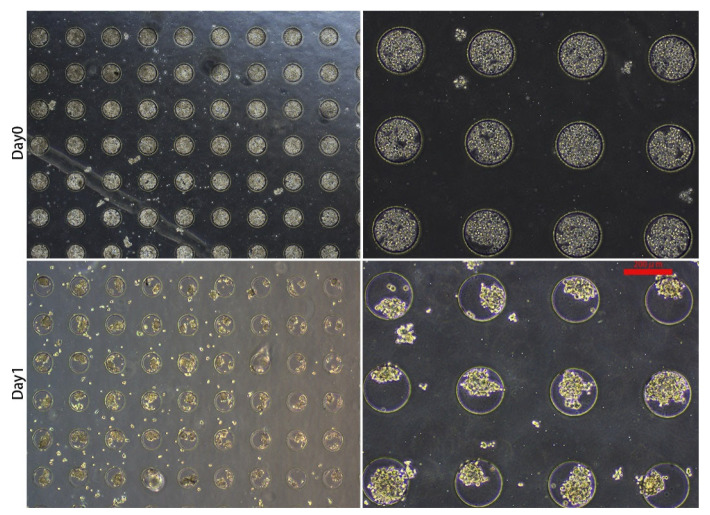
Morphology of SY-II cells on the day of seeding (Day 0) and spheroid formation (Day 1). On day 0, about 30 μL cell suspension with a concentration of 1 × 10^7^ cells/mL was added onto the chip with a pipette. On day 1, 3D spheroid formation occurred. Magnification = 40× (two pictures on the **left**). Magnification = 100× (two pictures on the **right**). Bar = 200 μm.

**Figure 4 micromachines-13-01645-f004:**
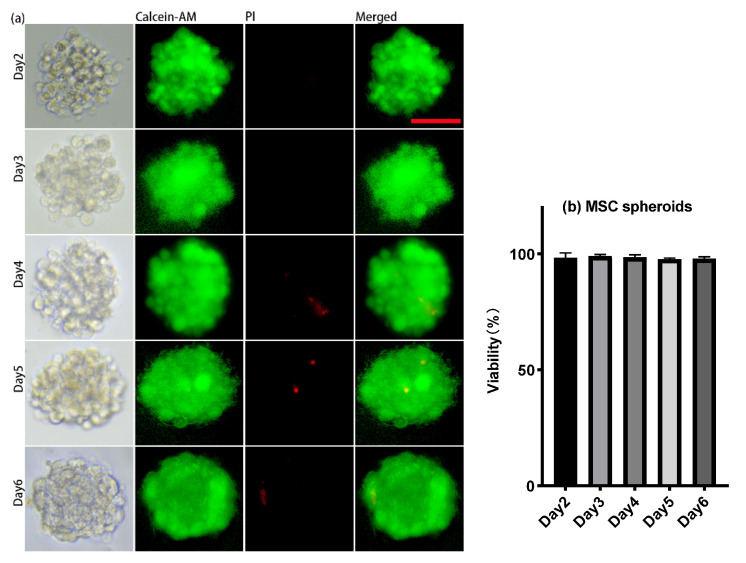
(**a**) Cell viability test of MSC spheroids: living cells stained with Calcein AM (green) and dead cells stained with PI (red). (**b**) Histograms showed the cell viability of MSC spheroids on day 2, day 3, day 4, day 5, and day 6. Magnification = 200×, Bar = 50 μm.

**Figure 5 micromachines-13-01645-f005:**
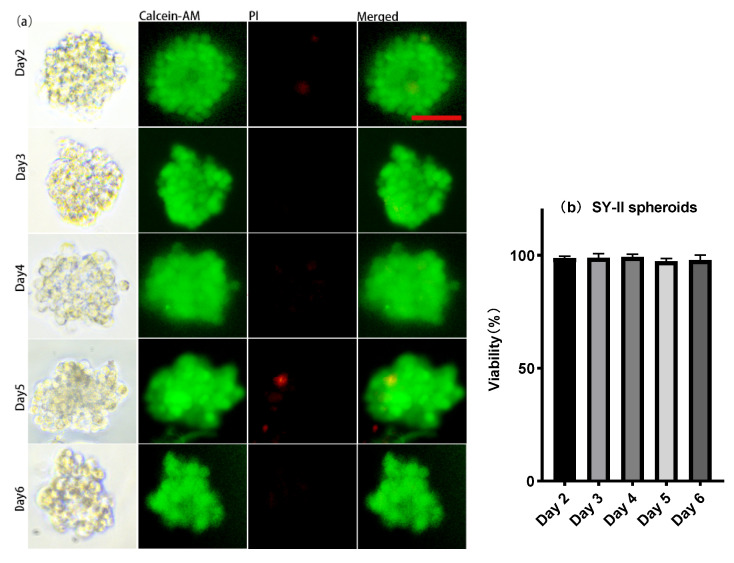
(**a**) Cell viability test of SY-II spheroids: living cells stained with Calcein AM (green) and dead cells stained with PI (red). (**b**) Histograms showed the cell viability of SY-II spheroids on day 2, day 3, day 4, day 5, and day 6. Magnification = 200×, Bar = 50 μm.

## Data Availability

Data sharing not applicable. No new data were created or analyzed in this study. Data sharing is not applicable to this article.

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
