# Peer review of "A High-Throughput and Uniform Amplification Method for Cell Spheroids"

_micromachines, 2022, doi:10.3390/mi13101645_

Round 1

Reviewer 1 Report

This manuscript describes a high-throughput and uniform amplification method for cell spheroids. The authors make use of a novel microporous chip to generate uniform spheres first followed by amplification of spheres in a conventional stirring culture flask. PDMS (polydimethylsiloxane) was used in the chip and the coating of the stirring flask.

Additional comments and requests:

Figure 1: Bar (400 µM) can be hardly seen, change the colour of the bar e.g. black (1h). In addition, add a bar to indicate the size of the chip (1i).

Figure 2-5: Bars can be hardly seen, change colour or make them bold.

Methods:

Description of the microporous chip should be more detailed: add size of the chip, number of microspores in a chip, size and volume of the microporous

How many cells settle down in one microporous to generate a sphere?

Results:

How many spheres were transferred into one stirring culture flask, and after cultivation of 5 days, how many spheroids were generated in total?

Reviewer 2 Report

The paper “A High-Throughput and Uniform Amplification Method for Cell Spheroids” by Liu et al presents an interesting way to form cell spheroids, yet it is overall unconvincing that the technology is more advance than prior works. It is recommended that the authors can better survey the literature and better articulate the advantages of the work. There are also other technical questions. It is recommended the article can be revised before publication.

1.    There are many prior works providing similar functions or even better capability. It is recommended that the authors can better survey the literature and better articulate the advantages of the work. Some relevant works are attached below:
"High-Throughput Cancer Cell Sphere Formation for Characterizing the Efficacy of Photo Dynamic Therapy in 3D Cell Cultures"
"Drug testing and flow cytometry analysis on a large number of uniform sized tumor spheroids using a microfluidic device"

2.    Given the authors can generate spheroids with a good throughput, the capability to test many treatment conditions has not been developed. It might not be really helpful for this research society. There are already existing platforms that can test combinations of many drugs. It is recommended that the authors can discuss that.

3.    It is not clear whether the "stirring culture flask" is placed in an incubator or not. If yes, is there a way to automatically stir it? Manual stirring is really inefficient. If not, how to provide a suitable temperature and CO2 control?

4.    While the authors show good viability of spheroids, The spheroids do not seem to grow. It is recommended that the authors can demonstrate growth curves of spheroids. If they are not growing, probably still dormant/unhealthy.

1.    3.1.3. “D” Spheroid Formation and Maintenance looks like a typo.

"

Round 2

Reviewer 2 Report

While the authors included two prior publications for comparison, it is recommended that the authors can include more relevant prior works for this application and discuss the pros and cons. Other than that, the paper can be published.
